# The prevalence of esophageal cancer after caustic and pesticide ingestion: A nationwide cohort study

**Han-Wei Mu**[1,2], **Chun-Hung Chen**[1,2], **Kai-Wei Yang**[1,2], **Chi-Syuan Pan**[1,2], **Cheng-Li Lin**[3], **Dong-Zong Hung**[1]*

1 Division of Toxicology, China Medical University Hospital, Taichung, Taiwan, 2 Department of Emergency Medicine, China Medical University Hospital, Taichung, Taiwan, 3 Management Office of Health Data, China Medical University Hospital, Taichung, Taiwan

* dzhung0224@gmail.com

**Data Availability Statement:** All relevant data are within the paper and its Supporting Information files.

## Abstract

Habits such as smoking and alcohol drinking and existing esophageal malfunction are considered the main risk factors for esophageal carcinogenesis. Caustic ingestion of acidic or alkaline agents or strong irritants can induce severe esophageal corrosive injury and increase esophageal cancer risk. We studied the relationship between esophageal carcinoma and acute detergent or pesticide poisoning by using nationwide health insurance data. **Methodology/Principle findings**: We compared a pesticide/detergent intoxication cohort (N = 21,840) and an age- and gender-matched control cohort (N = 21,840) identified from the National Health Insurance Research Database between 2000 and 2011. We used the multivariable Cox proportional model to determine esophageal carcinoma risk. The overall incidence density of esophageal cancer was 1.66 per 10,000 person-years in the comparison cohort and 4.36 per 10,000 person-years in the pesticide/detergent intoxication cohort. The corresponding adjusted hazard ratio (HR) for esophageal cancer was 2.33 (95% confidence interval [CI] = 1.41–3.86) in the pesticide/detergent intoxication cohort compared with the control cohort. Patients with corrosive and detergent intoxication did not have a higher risk of esophageal cancer (adjusted HR = 0.98, 95% CI = 0.29–3.33) than those without pesticide/detergent intoxication. However, patients with pesticide intoxication had a significantly higher risk of esophageal cancer (adjusted HR = 2.52, 95% CI = 1.52–4.18) than those without pesticide/detergent intoxication. **Conclusion**: In the present study, after adjusting for conventional risk factors, we observed that pesticide intoxication could exert substantial effects through increased esophageal cancer risk. However, patients with detergent intoxication may not have an increased risk of esophageal cancer.

## Introduction

Essentially, self-ingestion of caustic agents, detergents, and pesticides is a serious public health problem in Taiwan. According to the Taiwan health statistics, 600 people ingested liquid toxins, including caustic agents and pesticides, for suicidal attempt in 1 year, and this is the third

**Funding:** This work was supported by grants from the Ministry of Health and Welfare, Taiwan (MOHW108-TDU-B-212-133004), China Medical University Hospital (DMR-107-192, CMU107-ASIA-19), Academia Sinica Stroke Biosignature Project (BM10701010021), MOST Clinical Trial Consortium for Stroke (MOST 107-2321-B-039 -004-), Tseng-Lien Lin Foundation, Taichung, Taiwan, and Katsuzo and Kiyo Aoshima Memorial Funds, Japan. The funders had no role in the study design, data collection and analysis, decision to publish, or preparation of the manuscript. No additional external funding was received for this study.

**Competing interests:** The authors have declared that no competing interests exist.

common method for committing suicide. Caustic substance ingestion is most frequently encountered in children as a result of accidental swallowing or in adults as a result of self-harm. It often extensively injures the upper gastrointestinal tract and may lead to extensive necrosis, perforation, and death. Among the agents for pesticide poisoning in Taiwan, organo-phosphorus, herbicides, and other pesticides account for 45%, 23%, and 23%, respectively, according to the admission 2009 data from Taiwan National Health Insurance Database.

Esophageal cancer accounts for >500,000 cancer deaths annually, and the incidence is rapidly increasing worldwide [1]. In Taiwan, 2,630 new cases of esophageal cancer and 1,792 deaths caused by esophageal cancer occurred in 2013. The mean age at occurrence was 57 in men and 62 in women. Most esophageal cancer cases in Taiwan are of squamous cell carcinoma (93%), and the incidence is still increasing. The risk factors for esophageal cancer are smoking, alcohol consumption, dietary factors such as betel quid chewing and high temperature beverage consumption, gastroesophageal reflux disease, and underlying esophageal diseases such as achalasia, and they are substantially different in various parts of the world [1–5].

Some studies have shown that caustic ingestion that induced severe esophageal corrosive injury might increase esophageal cancer risk [6–12]. Some studies with limited data even estimated a 1,000-fold higher risk [6]. However, these studies were based on a small number of case control studies; hence, the evidence is not strong.

Pesticides protect plants from weeds, fungi, or insects. Pest control agents are usually applied through chemical dispersal in a hydrocarbon solvent-surfactant system to provide a homogeneous preparation. In addition to pesticides, these solvent-surfactants, such as the surfactant of glyphosate, produce significant mucosal irritation effects. Some epidemiological studies have demonstrated high risks of certain cancers from exposure to some solvents [13]. Some pesticides are classified as carcinogenic or potentially carcinogenic to humans, such as captafol, diazinon, malathion, and glyphosate. Here, our study investigated the relationship between esophageal cancer and esophageal injuries after caustics ingestion and pesticide poisoning.

## Methods

### Data source

This study used data from the National Health Insurance Research Database (NHIRD). The NHIRD was launched in Taiwan in 1995 and covers nearly 99% of the total population of Taiwan with comprehensive healthcare benefits. For this study, we used the deidentified data of the residents to link two data files (subsets of the NHIRD), namely inpatient claims data and Registry of Beneficiaries. International Classification of Diseases-9-Clinical Modification (ICD-9-CM) codes were used to define diseases in the NHIRD. This study was approved by the Ethics Review Board of China Medical University (CMUH-104-REC2-115).

### Study population

Patients with pesticide/detergent intoxication were identified from the NHIRD from January 1, 2000, to December 31, 2005, according to ICD-9-CM codes 983, 989.3–989.4, and 989.6. Patients diagnosed with cancer (ICD-9-CM codes 140–208) before pesticide/detergent intoxication or those who lacked continuous health insurance coverage preceding cohort entry were excluded. Furthermore, all patients aged <20 years were excluded. Moreover, the comparison cohort of individuals without any history of pesticide/detergent intoxication was identified from the NHIRD. The comparison cohort also excluded those with cancer history, without health insurance before entering the study, or aged <20 years. In the final cohort, the pesticide/detergent intoxication cohort was matched to the comparison cohort at a 1:1 ratio by

gender, age, and the year of study entry. We designated 50 and 65 years as the age threshold. A consensus is lacking regarding the age at which an individual can be considered elderly, but the World Health Organization defines individuals >65 years as elderly in most developed countries. In less developed countries, for example in parts of Africa, >50 years old is considered elderly. Thus, we classified participants into the age groups of <49, 50–64, and >65 to determine the difference between each group.

The index date was defined as the date of first diagnosis of pesticide/detergent intoxication in the database. All participants were observed until they were diagnosed with esophageal cancer (ICD-9-CM code 150), death, or the end of the study period (December 31, 2011).

## Outcome, comorbidity, and medication

The primary clinical outcome was esophageal cancer (ICD-9-CM code 150). Furthermore, participants in the pesticide/detergent intoxication and control cohorts were compared for common comorbidities, including hypertension (ICD-9-CM codes 401–405), diabetes mellitus (ICD-9-CM code 250), chronic obstructive pulmonary disease (ICD-9-CM codes 491, 492, and 496), obesity (ICD-9-CM code 278), alcohol-related illness (ICD-9-CM codes 291, 303, 305, 571.0, 571.1, 571.2, 571.3, 790.3, A215, and V11.3), ischemic heart disease (ICD-9-CM codes 410–414), cerebrovascular disease (ICD-9-CM codes 430–438), and gastric disease (ICD-9-CM codes 530–534). Common comorbidities were identified according to the diagnosis records in the inpatient file before the index date.

## Statistical analysis

We used descriptive statistics to summarize the characteristics of the pesticide/detergent intoxication cohort and matched comparison cohort. A continuous variable, such as age, was used in an independent *t* test to examine the mean ages between the two cohorts. Categorical variables are presented as the number and percentage and included sex and common comorbidity assessed using the chi-square test. Univariable and multivariable Cox proportional hazard regression analyses were used to determine esophageal cancer risk, and the results are presented as hazard ratios (HRs) with 95% confidence intervals (CIs). The differences in the cumulative incidence of esophageal cancer between the pesticide/detergent intoxication and control cohorts were estimated using the Kaplan–Meier method with the log-rank test. A two-tailed p value of <0.05 was considered statistically significant. We used SAS software (version 9.4 for windows; SAS Institute, Cary, NC, USA) for all statistical analyses and Kaplan–Meier survival curve plots.

## Results

This study included 21,840 patients with pesticide/detergent intoxication and 21,840 control patients. The basic characteristics of the two cohorts are shown in Table 1. The mean ages of the pesticide/detergent intoxication cohort and comparison cohort were 52.1 ± 17.4 and 51.6 ± 17.6, respectively. No significant difference was noted in sex and age. The majority of pesticide/detergent intoxication patients were men (62.1%) and <49 years old (48.1%). In general, a high proportion of pesticide/detergent intoxication patients had hypertension, diabetes mellitus, gastric disease, ischemic heart disease, cerebrovascular disease, chronic obstructive pulmonary disease, alcohol-related illness, and obesity (all p < 0.001). The average follow-up duration was 5.25 ± 3.86 years for the pesticide/detergent cohort and 6.63 ± 3.29 years for the comparison cohort. The Kaplan–Meier curve showed that the cumulative incidence of esophageal cancer was higher in the pesticide/detergent cohort than in the comparison cohort throughout the 12-year follow-up period (Fig 1). The cumulative incidence of esophagus

**Table 1. Characteristics of patients with and without pesticide/detergent intoxication.**

| | Pesticide/Detergent intoxication | | | | | |
| --- | --- | --- | --- | --- | --- |
| | Yes | | No | | |
| | (N = 21840) | | (N = 21840) | | |
| | n | % | n | % | p-value |
| Age, year | | | | | 0.99 |
| ≤49 | 10496 | 48.1 | 10496 | 48.1 | |
| 50–64 | 5386 | 24.7 | 5386 | 24.7 | |
| ≥ 65 | 5958 | 27.3 | 5958 | 27.3 | |
| Mean (SD) [#] | 52.1 | 17.4 | 51.6 | 17.6 | 0.004 |
| Gender | | | | | 0.99 |
| Female | 8269 | 37.9 | 8269 | 37.9 | |
| Male | 13571 | 62.1 | 13571 | 62.1 | |
| Comorbidity | | | | | |
| Hypertension | 3454 | 15.8 | 1870 | 8.56 | <0.001 |
| Diabetes mellitus | 2136 | 9.78 | 957 | 4.38 | <0.001 |
| Gastric disease | 3027 | 13.9 | 1057 | 4.84 | <0.001 |
| Ischemic heart disease | 1783 | 8.16 | 862 | 3.95 | <0.001 |
| Cerebrovascular disease | 1875 | 8.59 | 934 | 3.82 | <0.001 |
| Chronic obstructive pulmonary disease | 1044 | 4.78 | 424 | 1.94 | <0.001 |
| Alcohol-related illness | 1071 | 4.90 | 112 | 0.51 | <0.001 |
| Obesity | 16 | 0.07 | 6 | 0.03 | <0.001 |

Chi-square test.

[#] $t$ test.

cancer was significantly different between the pesticide/detergent and comparison cohorts (log-rank test; p < 0.001).

The overall incidence densities of esophageal cancer were 1.66 and 4.36 per 10,000 person-years in the comparison and pesticide/detergent cohorts, respectively (Table 2). The corresponding adjusted HR for esophageal cancer was 2.33 (95% CI = 1.41–3.86) compared with controls after adjusting for age, sex, gastric disease, and alcohol-related illness. Compared with patients aged <49 years, those aged 50–64 and >65 years had 2.67-fold (95% CI = 1.54–4.64) and 3.18-fold (95% CI = 1.74–5.80) significantly higher risks of esophageal cancer, respectively. Compared with women, men had an adjusted HR of 19.8 (95% CI = 4.85–80.8) for esophagus cancer. Among various comorbidity types, significantly increased risk was observed in those with alcohol-related illness (adjusted HR = 7.14, 95% CI = 3.63–14.1).

Table 3 presents the incidence and HR of esophageal cancer between patients with and without pesticide/detergent intoxication. Compared with patients without pesticide/detergent intoxication, men, patients aged <49 years, and those aged >65 years with pesticide/detergent intoxication had 2.22-fold (95% CI = 1.34–3.69), 2.84-fold (95% CI = 1.08–7.47), and 2.94-fold (95% CI = 1.19–7.26) increased esophagus cancer risks, respectively. For patients without comorbidity, those with pesticide/detergent intoxication had a significantly higher esophageal cancer risk than those without pesticide/detergent intoxication (adjusted HR = 2.32, 95% CI = 1.32–4.10). Among patients with non–alcohol-related illness, those with pesticide/detergent intoxication had a higher risk of esophageal cancer than controls (adjusted HR = 2.47, 95% CI = 1.46–4.16).

Table 4 presents the incidence and adjusted HR of esophageal cancer between different groups of patients with pesticide/detergent intoxication. Patients with corrosive and detergent

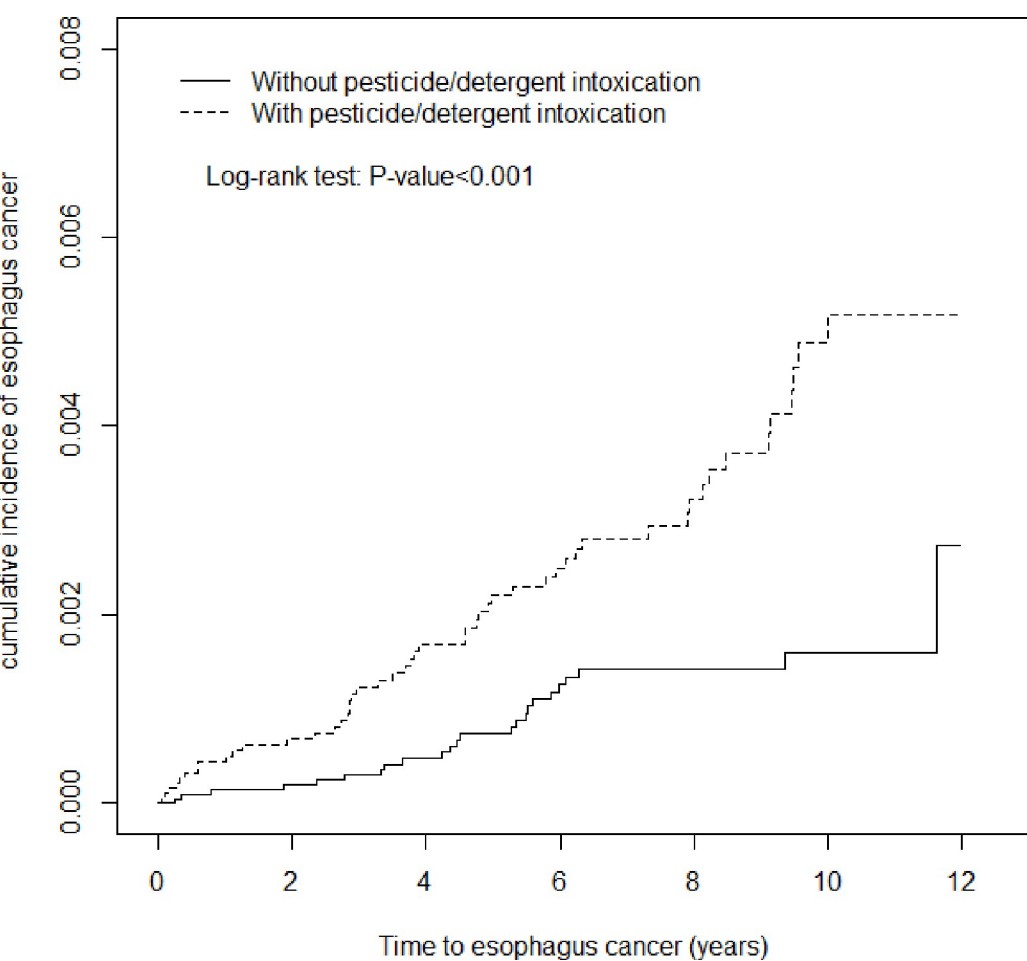

**Fig 1.** Cummulative incidence comparison of esophagus cancer in patients with (dashed line) and without (solid line) pesticide/detergent intoxication.

intoxication (ICD-9-CM codes 983 and 989.6) did not have a higher risk of esophageal cancer (adjusted HR = 0.98, 95% CI = 0.29–3.33) than those without pesticide/detergent intoxication. Furthermore, patients with only pesticide intoxication (ICD-9-CM codes 989.3 and 989.4) had a significantly higher risk of esophageal cancer (adjusted HR = 2.52, 95% CI = 1.52–4.18) than those without pesticide/detergent intoxication.

## Discussion

Several factors, including living habits and hobbies, contribute to esophageal cancer development. Esophageal cancer has two major subtypes, namely squamous cell carcinoma and adenocarcinoma, which have some same and different risk factors. Several genetic and epigenetic alterations are implicated in both the development and progression of esophageal cancer. Mucosal break, inflammation, and toxic injuries caused by excessive alcohol drinking and heavy smoking, two of the most important and common risk factors, are causes of esophageal carcinoma. Although the relationship between caustic ingestion and esophageal cancer and the mechanism of esophageal cancer development are unclear, lye-based cleaner burn has been found to complicate esophageal strictures and thus increase the risk of esophageal squamous cell carcinoma [6,7,11]. Fewer case series studies have shown that the esophageal cancer

**Table 2. Incidence per 10,000 person-years of and risk factors for esophagus cancer.**

| Variable | Event | PY | Rate# | Crude HR (95% CI) | Adjusted HR& (95% CI) |
|---|---|---|---|---|---|
| Pesticide/Detergent intoxication | | | | | |
| No | 24 | 144761 | 1.66 | 1.00 | 1.00 |
| Yes | 50 | 114723 | 4.36 | 2.64(1.63, 4.30)*** | 2.33(1.41, 3.86)** |
| Age, year | | | | | |
| ≤49 | 22 | 137357 | 1.60 | 1.00 | 1.00 |
| 50–64 | 30 | 66141 | 4.54 | 2.85(1.65, 4.95)*** | 2.67(1.54, 4.64)*** |
| ≥ 65 | 22 | 55986 | 3.93 | 2.55(1.41, 4.61)** | 3.18(1.74, 5.80)*** |
| Gender | | | | | |
| Female | 2 | 100714 | 0.20 | 1.00 | 1.00 |
| Male | 72 | 158769 | 4.53 | 22.9(5.63, 93.5)*** | 19.8(4.85, 80.8)*** |
| Comorbidity | | | | | |
| Hypertension | | | | | |
| No | 66 | 238183 | 2.77 | 1.00 | 1.00 |
| Yes | 8 | 21300 | 3.76 | 1.41(0.68, 2.95) | |
| Diabetes mellitus | | | | | |
| No | 68 | 247603 | 2.75 | 1.00 | 1.00 |
| Yes | 6 | 11880 | 5.05 | 1.92(0.83, 4.42) | |
| Gastric disease | | | | | |
| No | 64 | 242170 | 2.64 | 1.00 | 1.00 |
| Yes | 10 | 17314 | 5.78 | 2.25(1.15, 4.38)* | 0.83(0.40, 1.72) |
| Ischemic heart disease | | | | | |
| No | 71 | 248574 | 2.86 | 1.00 | 1.00 |
| Yes | 3 | 10910 | 2.75 | 0.99(0.31, 3.15) | |
| Cerebrovascular disease | | | | | |
| No | 71 | 248837 | 2.85 | 1.00 | 1.00 |
| Yes | 3 | 10646 | 2.82 | 1.02(0.32, 3.25) | |
| Chronic obstructive pulmonary disease | | | | | |
| No | 71 | 254217 | 2.79 | 1.00 | 1.00 |
| Yes | 3 | 5266 | 5.70 | 2.13(0.67, 6.77) | |
| Alcohol-related illness | | | | | |
| No | 61 | 254356 | 2.40 | 1.00 | 1.00 |
| Yes | 13 | 5128 | 25.4 | 10.9(5.96, 19.8)*** | 7.14(3.63, 14.1)*** |
| Obesity | | | | | |
| No | 74 | 259400 | 2.85 | 1.00 | 1.00 |
| Yes | 0 | 83 | 0.00 | - | |

Rate#: incidence rate per 10,000 person-years.

Crude HR, relative hazard ratio.

Adjusted HR&: Multivariable analysis including age, sex, gastric disease, and alcohol-related illness.

*p < 0.05

**p < 0.01

***p < 0.001.

incidence caused by caustic ingestion is 1.4%–2.6% [6,7,14]. Although the incidence might be overestimated, most experts agree that corrosive injury might be a risk factor for esophageal carcinoma and have even alleged that the risk is 1,000 times that in the general population [15]. However, the results of this study are very different from those in the literature. This research is a nationwide, population-based cohort study designed to identify whether a

**Table 3. Incidence and hazard ratio of esophageal cancer between patients with and without pesticide/detergent intoxication.**

| Variables | Pesticide/Detergent intoxication | | | | | | Crude HR (95% CI) | Adjusted HR[&] (95% CI) |
|---|---|---|---|---|---|---|---|---|
| | No | | | Yes | | | | |
| | Event | PY | Rate[#] | Event | PY | Rate[#] | | |
| Gender | | | | | | | | |
| Female | 0 | 55515 | 0.00 | 2 | 45199 | 0.44 | - | - |
| Male | 24 | 89246 | 2.69 | 48 | 69523 | 6.90 | 2.59(1.59, 4.23)*** | 2.22(1.34, 3.69)** |
| Age, year | | | | | | | | |
| $\leq 49$ | 6 | 74309 | 0.81 | 16 | 63048 | 2.54 | 3.16(1.24, 8.08)* | 2.84(1.08, 7.47)* |
| 50–64 | 11 | 36962 | 2.98 | 19 | 29178 | 6.51 | 2.23(1.06, 4.68)* | 1.63(0.74, 3.57) |
| $\geq 65$ | 7 | 33489 | 2.09 | 15 | 22496 | 6.67 | 3.19(1.30, 7.83)* | 2.94(1.19, 7.26)* |
| Comorbidity | | | | | | | | |
| No | 20 | 130000 | 1.54 | 30 | 90783 | 3.30 | 2.16(1.22, 3.80)** | 2.32(1.32, 4.10)** |
| Yes | 4 | 14760 | 2.71 | 20 | 23939 | 8.35 | 3.07(1.05, 8.98)* | 2.77(0.92, 8.31) |
| Alcohol-related illness | | | | | | | | |
| No | 22 | 144121 | 1.53 | 39 | 110234 | 3.54 | 2.33(1.38, 3.94)*** | 2.47(1.46, 4.16)*** |
| Yes | 2 | 639 | 31.3 | 11 | 4489 | 24.5 | 0.78(0.17, 3.53) | 1.26(0.27, 5.94) |

PY, person-years.

Rate[#]: incidence rate per 10,000 person-years.

Crude HR, relative hazard ratio.

Adjusted HR[†]: Multivariable analysis including age, sex, gastric disease, and alcohol-related illness.

*p < 0.05

**p < 0.01

***p < 0.001.

significant association exists between caustic ingestion and the risk of subsequent esophageal cancer. We defined conventional risk factors for esophageal cancer, such as age, sex, smoking, alcohol abuse, and gastric disease (such as achalasia and GERD), which were already well-established previously. In this 1-million-people cohort, 4,429 people were included in the detergent and corrosive intoxication group. The relative risk of esophageal cancer did not increase in patients with caustic agent and detergent poisoning compared with those without the poisoning after adjustment for these conventional risk factors. One of the reasons might be that our study included patients with exposure to detergents with less caustic characteristics. Detergents with acidic or alkaline characteristics are some of the most used toxic and corrosive

**Table 4. Incidence and adjusted hazard ratio of esophageal cancer between different entities of pesticide/detergent intoxication.**

| Variable | N | No. of Events | Rate[#] | Adjusted HR[†] | 95% CI |
|---|---|---|---|---|---|
| Without Pesticide/Detergent intoxication | 21840 | 24 | 1.66 | 1.00 | (Reference) |
| With Organophosphate/Carbamate + Pesticide (ICD-9-CM code 989.3, 989.4) | 17411 | 47 | 5.31 | 2.52 | (1.52, 4.18) |
| With Detergent (ICD-9-CM code 983, 989.6) | 4429 | 3 | 1.14 | 0.98 | (0.29, 3.33) |

PY, person-years.

Rate[#], incidence rate per 10,000 person-years.

Crude HR, relative hazard ratio.

Adjusted HR[†]: Multivariable analysis including age, sex, gastric disease, and alcohol-related illness.

*p < 0.05, **p < 0.01, ***p < 0.001.

chemicals at home. In general, detergents are classified into three categories according to their surfactant electrical charge: nonionic, anionic, and cationic. Nonionic and anionic detergents have low toxicity, although they may be mild to moderate irritants. Most serious toxins are cationic detergents. Most of the detergents used at home are nonionic and anionic. Therefore, patients with ICD-9-CM codes 983.1 and 983.2 (acidic and alkali corrosive injury), 983.9 (caustic intoxication), or 989.6 (detergent intoxication) were identified, which expanded the dataset and weakened the results. The grade of esophagus corrosive injury of these cases is not available in the database. Thus, the true risk of esophageal cancer might be underestimated because, theoretically, esophageal cancer commonly occurs in patients with high-grade esophageal corrosive injury. However, based on these data, the results still have considerable credibility.

Another reason might be that the exposure interval after intoxication is shorter in this study than in previous studies (only 12 years with an average follow-up duration of only 5 years more). The results might be different if we increased the data and extended the study period. In previous studies, lye ingestion resulted in squamous cell carcinoma in the esophagus rather than adenocarcinoma [16]. Despite its uncertain etiology and pathogenesis, the mechanism of esophageal cancer after caustic agent and pesticide ingestion is probably similar to that of achalasia or esophageal diverticulum. The severe injury of esophagus after caustic ingestion causes lumen stricture or decreased esophageal motility. Subsequently, esophageal stasis occurs, which leads to local chronic inflammatory responses in the esophageal mucosa, which can lead to carcinogenesis. In cases of chronic irritation caused by foods and gastric fluid in achalasia, reflux esophagitis, or Barrett's esophagus, the interval between disease diagnosis and esophageal carcinoma development was approximately 10–15 years [17]. However, the interval was considered to be shortened to 4 years for patients with aforementioned diseases who were exposed to airborne toxins that resulted from the terrorist attack of the World Trade Center [18]. Chemical hazard exposure can accelerate solid tumor development, such as esophageal carcinoma. In total, 287 chemicals or chemical groups with potential carcinogenic effects were identified in the field of the World Trade Center, including several organic solvents used in pesticide synthesis.

In this study, the relative risk of esophageal cancer increased significantly by 2.52× in the pesticide group, and it was 2.47× even after excluding the comorbidity of alcohol-related illness. Some pesticides are considered to become carcinogenic over a long time, including their main ingredients or organic solvents. However, such carcinogenicity was identified for most of them after chronic exposure in in vitro, in vivo, or epidemiological studies. No study has examined the relationship of acute large dose exposure with the occurrence of esophageal cancer. However, some studies have reported that esophageal cancer is positively associated with intensive pesticide exposure. Jansson et al. found increased esophageal adenocarcinoma risk among people with high exposure to pesticides [19]. Meyer et al. showed that esophageal cancer is correlated with pesticide exposure because of the high mortality caused by esophageal cancer in states in Brazil using a high proportion of pesticides [20]. Several pesticides have been identified as carcinogens, including their main ingredients or solvents. Animal studies have demonstrated strong genotoxicity for some pesticides, such as diazinon organophosphates, malathion, and glyphosate herbicide, due to DNA and chromosomal damage. Furthermore, numerous animal studies have shown strong cellular oxidative stress reactions for them. Glyphosate herbicide damages the retro-pharynx and esophagus more severely than other pesticides and causes a high rate of morbidity among patients because of its surfactant (polyethoxylated tallowamine) [21,22]. In this cohort study, a high proportion of patients in the pesticide/detergent intoxication cohort had hypertension, diabetes mellitus, gastric disease, ischemic heart disease, cerebrovascular disease, chronic obstructive pulmonary disease,

alcohol-relative illness, and obesity (all p < 0.001). Single severe direct esophageal mucosa damage as well as subsequent inflammation might be one of the causes of carcinogenicity in these patients with chronic systemic diseases and on long-term medication, with possible malfunction of the esophagus and stomach. However, in-depth animal experiments and studies are required to explore the possible mechanisms of the correlation.

Our study has several limitations. The data were collected based on the ICD-9-CM codes in the database; therefore, some detailed information could not be obtained. First, the grade of esophageal corrosive injury after caustic ingestion is not provided in the database. This might underestimate the true risk of esophageal cancer because, theoretically, esophageal cancer occurs commonly in patients with high-severity esophageal corrosive injury. Second, although ICD-9-CM codes are used for acidic and alkali corrosive injury (983.1 and 983.2), most doctors in Taiwan refer such patients for the diagnosis of caustic intoxication (ICD-9-CM code 983.9) or detergent intoxication (ICD-9-CM code 989.6). It makes a huge difference in the case numbers between these diagnoses. Therefore, we cannot evaluate esophageal cancer risk in patients with acidic and alkaline caustic injury accurately. Third, although caustic, detergent, and pesticide intoxication in Taiwan are mostly through the oral route, using a diagnostic code to represent all oral-route intoxication could still slightly affect the results. Fourth, due to the limitation of the ICD-9-CM diagnostic codes, we could categorize the pesticides used for further detailed analysis. Furthermore, we were unable to extract the exact pathology reports from the database; thus, we could not further categorize the pathologies into premalignancy lesions, such as polyp or hyperplasia, or malignancies, such as adenocarcinoma or squamous cell carcinoma. Fifth, because a health insurance claims database was used, detailed information on certain general characteristics, such as obesity, body mass index, smoking, exercise, and dietary habits, was lacking. To compensate, we tried to use clinical examination-related morbidities to correct the individual examination index. Lastly, the present research involved only the Taiwanese general population, which includes 99.5% Han Chinese; thus, differences may be apparent in a stratified population.

Conclusively, to determine the association between corrosive and detergent intoxication and esophageal cancer risk, the present study analyzed a population-based cohort from a nationwide claims database and adjusted for comorbidities to comprehensively assess corrosive intoxication-related esophageal cancer risk. We observed that patients with preexisting corrosive poisoning did not exhibit a higher esophageal cancer risk than the general population. However, preexisting pesticide intoxication was associated with a 2.5-fold higher risk of esophageal cancer compared with the general population. Further investigations are required to delineate the association between esophageal carcinoma and esophageal corrosive injury or pesticide poisoning.

## Author Contributions

**Conceptualization:** Han-Wei Mu, Chi-Syuan Pan.

**Data curation:** Cheng-Li Lin.

**Formal analysis:** Cheng-Li Lin.

**Methodology:** Cheng-Li Lin.

**Supervision:** Chun-Hung Chen, Dong-Zong Hung.

**Writing – original draft:** Han-Wei Mu.

**Writing – review & editing:** Kai-Wei Yang, Dong-Zong Hung.

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
