## [Decision Letter · Decision Letter 0]

28 Aug 2020

PONE-D-20-20245

The prevalence of esophageal cancer after caustic and pesticide ingestion: A nationwide cohort study

PLOS ONE

Dear Dr. Hung,

Thank you for submitting your manuscript to PLOS ONE. After careful consideration, we feel that it has merit but does not fully meet PLOS ONE’s publication criteria as it currently stands. Therefore, we invite you to submit a revised version of the manuscript that addresses all the points raised during the review process.

We look forward to receiving your revised manuscript.

Kind regards,

Gianluigi Forloni

Academic Editor

PLOS ONE

Journal Requirements:

2.We suggest you thoroughly copyedit your manuscript for language usage, spelling, and grammar. If you do not know anyone who can help you do this, you may wish to consider employing a professional scientific editing service.  

3.Thank you for stating the following in the Acknowledgments Section of your manuscript:

[This study is supported in part by Taiwan Ministry of Health and Welfare Clinical Trial

Center (MOHW109-TDU-B-212-114004), MOST Clinical Trial Consortium for Stroke

(MOST 108-2321-B-039-003-), Tseng-Lien Lin Foundation, Taichung, Taiwan.]

 [The authors received no specific funding for this work.]

5. Please upload a copy of Figure 1, to which you refer in your text on page 8. If the figure is no longer to be included as part of the submission please remove all reference to it within the text.

Reviewers' comments:

Reviewer's Responses to Questions

**Comments to the Author**

1. Is the manuscript technically sound, and do the data support the conclusions?

Reviewer #1: Yes

2. Has the statistical analysis been performed appropriately and rigorously? 

Reviewer #1: Yes

3. Have the authors made all data underlying the findings in their manuscript fully available?

Reviewer #1: Yes

4. Is the manuscript presented in an intelligible fashion and written in standard English?

Reviewer #1: Yes

5. Review Comments to the Author

Reviewer #1: n the current study, Mu et al. explored the relationship of the occurrence of esophageal carcinoma and acute detergent or pesticide poisoning based on a nationwide Health Insurance Data. This work is within the scope of the journal. The authors used the English language professionally throughout the manuscript and the manuscript was appropriately structured. Here, I have pointed out minor points.

1. It is not clear how the threshold of age (≤49,50-64,≥65) is obtained in the work.

2.Among several co-morbidities, significant elevated risk was observed in those with alcohol-related illness in current study. In order to strengthen the results, the relationships between incidence of esophagus cancer and various variables should also be analyzed in subgroups with and without alcohol-related illness respectively.

3.As shown in table 4, patients who only with pesticide intoxication had significant higher risk of esophagus cancer. Thus, all the analyses should be repeated in three subgroups: pesticide intoxication, detergent intoxication and without pesticide/ detergent intoxication

4.As we known, the formation of esophagus cancer is a long process. Therefore, several associated clinicopathological factors (e.g. esophagitis, epithelial hyperplasia, polyp) should also be taken into consideration.

4. The references are a bit old and more recent ones should be quoted.

5. Several grammatical errors need to be corrected.

6. PLOS authors have the option to publish the peer review history of their article (what does this mean?). If published, this will include your full peer review and any attached files.

Reviewer #1: No

---

## [Author Response · Author response to Decision Letter 0]

20 Oct 2020

1. It is not clear how the threshold of age (≤49,50-64,≥65) is obtained in the work.

Response:

We thank the reviewer for pointing this out. According to the World Health Organization (WHO) (http://www.who.int/healthinfo/survey/ageingdefnolder/en/) mentioned most developed world countries have accepted the chronological age of 65 years as a definition of 'elderly' or older person. The more traditional African definitions of an elder or 'elderly' person correlate with the chronological ages of 50 to 65 years, depending on the setting, the region and the country. So, we classified groups as below 49, 50-64 and above 65 and tried to find the difference between different classified groups.

We have revised the text of “Methods” as follows.

“In the final cohort, the pesticide/detergent intoxication cohort was matched to the comparison cohort at a 1:1 ratio by gender, age, and the year of study entry. We designated 50 and 65 years as the age threshold. A consensus is lacking regarding the age at which an individual can be considered elderly, but the World Health Organization defines individuals >65 years as elderly in most developed countries. In less developed countries, for example in parts of Africa, >50 years old is considered elderly. Thus, we classified participants into the age groups of <49, 50–64, and >65 to determine the difference between each group.”

2. Among several co-morbidities, significant elevated risk was observed in those with alcohol-related illness in current study. In order to strengthen the results, the relationships between incidence of esophagus cancer and various variables should also be analyzed in subgroups with and without alcohol-related illness respectively.

Response:

Thank you for your comments. In response to this comment, we have revised the Table 3 and revised the text as follows. 

“Patients with pesticide/detergent intoxication were significantly associated with increased risk of esophagus cancer than the patients without pesticide/detergent intoxication for without co-morbidity (adjusted HR=2.32, 95%CI=1.32-4.10). Among the non-alcohol-related illness subjects, patients with pesticide/detergent intoxication had even higher risk of esophagus cancer compared to the comparison cohort (adjusted HR = 2.47, 95% CI=1.46-4.16).”

3. As shown in table 4, patients who only with pesticide intoxication had significant higher risk of esophagus cancer. Thus, all the analyses should be repeated in three subgroups: pesticide intoxication, detergent intoxication and without pesticide/ detergent intoxication

Response:

We agreed with the reviewer. We believed that analyses between these three subgroups would be helpful. However, in this study, esophageal cancer predominately occurred in the pesticide group. There were only 3 patients in the detergent group, so we were unable to perform subgroup analysis. 

4. As we know, the formation of esophagus cancer is a long process. Therefore, several associated clinicopathological factors (e.g. esophagitis, epithelial hyperplasia, polyp) should also be taken into consideration.

Response:

We agreed with the reviewer. 

However, the study method of this cohort is to collect cases of the diagnostic code from the health insurance database. We were unable to extract the exact clinicopathological factors you mentioned from the database. Thanks for your reminder. This is also one of the limitations of this article. Thus, we revised text as follow,

“Fourth, due to the limitation of the ICD-9-CM diagnostic codes, we could categorize the pesticides used for further detailed analysis. Furthermore, we were unable to extract the exact pathology reports from the database; thus, we could not further categorize the pathologies into premalignancy lesions, such as polyp or hyperplasia, or malignancies, such as adenocarcinoma or squamous cell carcinoma.”

5. The references are a bit old and more recent ones should be quoted.

Response:

Thank you for the recommendation. We had changed some new references.

6. Several grammatical errors need to be corrected.

Response:

We had tried to correct them. And we have used professional scientific editing service to correct them.

---

## [Editor Report · Decision Letter 1]

1 Dec 2020

The prevalence of esophageal cancer after caustic and pesticide ingestion: A nationwide cohort study

PONE-D-20-20245R1

Dear Dr. Hung,

We’re pleased to inform you that your manuscript has been judged scientifically suitable for publication and will be formally accepted for publication once it meets all outstanding technical requirements.

Kind regards,

Gianluigi Forloni

Academic Editor

PLOS ONE
---

## [Editor Report · Acceptance letter]

14 Dec 2020

PONE-D-20-20245R1 

The Prevalence of Esophageal cancer after Caustic and Pesticide Ingestion: A Nationwide Cohort Study 

Dear Dr. Hung:

I'm pleased to inform you that your manuscript has been deemed suitable for publication in PLOS ONE. Congratulations! Your manuscript is now with our production department. 

Kind regards, 

on behalf of

Dr. Gianluigi Forloni 

Academic Editor

PLOS ONE